# Radiosurgery for Brain Metastases: Challenges in Imaging Interpretation after Treatment

**DOI:** 10.3390/cancers15205092

**Published:** 2023-10-21

**Authors:** Andrea Romano, Giulia Moltoni, Antonella Blandino, Serena Palizzi, Allegra Romano, Giulia de Rosa, Lara De Blasi Palma, Cristiana Monopoli, Alessia Guarnera, Giuseppe Minniti, Alessandro Bozzao

**Affiliations:** 1NESMOS Department, U.O.C. Neuroradiology “Sant’Andrea” University Hospital, 00189 Rome, Italy; andrea.romano@uniroma1.it (A.R.); giulia.moltoni@uniroma1.it (G.M.); blandinoantonella7@gmail.com (A.B.); serenapalizzi@gmail.com (S.P.); allecarletta@gmail.com (A.R.); giulia.derosa@uniroma1.it (G.d.R.); lara.deblasi93@gmail.com (L.D.B.P.); crimonopoli@gmail.com (C.M.); guarneraalessia@gmail.com (A.G.); alessandro.bozzao@uniroma1.it (A.B.); 2Department of Radiological, Oncological and Pathological Sciences, “Sapienza” University of Rome, 00138 Rome, Italy; 3IRCCS Neuromed, 86077 Pozzilli, Italy

**Keywords:** stereotactic radiosurgery, brain metastasis, magnetic resonance imaging, immunotherapy, radiation necrosis

## Abstract

**Simple Summary:**

Stereotactic radiosurgery is one of the main treatments for patients with brain metastases, with important achievements in terms of patient prognosis. Nevertheless, radiation treatment may induce brain changes, visible in neuroimaging, that are often difficult to distinguish from progressive disease. This review aims to provide an update on the innovative neuroimaging modalities to study brain changes after stereotactic radiosurgery, focusing on the differential diagnosis between the presence of disease and post-treatment effects.

**Abstract:**

Stereotactic radiosurgery (SRS) has transformed the management of brain metastases by achieving local tumor control, reducing toxicity, and minimizing the need for whole-brain radiation therapy (WBRT). This review specifically investigates radiation-induced changes in patients treated for metastasis, highlighting the crucial role of magnetic resonance imaging (MRI) in the evaluation of treatment response, both at very early and late stages. The primary objective of the review is to evaluate the most effective imaging techniques for assessing radiation-induced changes and distinguishing them from tumor growth. The limitations of conventional imaging methods, which rely on size measurements, dimensional criteria, and contrast enhancement patterns, are critically evaluated. In addition, it has been investigated the potential of advanced imaging modalities to offer a more precise and comprehensive evaluation of treatment response. Finally, an overview of the relevant literature concerning the interpretation of brain changes in patients undergoing immunotherapies is provided.

## 1. Introduction

Stereotactic radiosurgery (SRS) often represents the primary modality of choice in the treatment of intact brain metastases (BM), or it can be used as adjuvant treatment after surgical resection [1]. The role of SRS in BM treatment has significantly evolved, and its goal is not only to locally control tumoral growth but also to delay or avoid whole brain radiation therapy (WBRT), reducing the incidence of toxicity radio-induced delay [1,2,3].

MRI is the gold standard for the detection of BM and for studying their evolution after SRS [4]. Several imaging techniques are nowadays available for monitoring the response to treatment after brain irradiation, ranging from basic MR examinations to more advanced MR techniques, such as perfusion studies [5], up to nuclear medicine examinations [6]. Nevertheless, response assessment after radiation therapy (RT) remains challenging, and the question to be answered is if there are tools that can indicate if there has been or not a response to therapy and if there are post-treatment changes mimicking tumoral disease.

According to RANO criteria, the response to the therapy is based mostly on dimensional criteria, with a complete response if lesions disappear, a partial response with at least a 30% decrease in the sum longest diameter of Central Nervous System (CNS) target lesions, progressive disease with at least a 20% increase in the sum longest diameter of CNS target lesions, and stable disease if dimensional variation does not qualify for partial response nor progressive disease [7,8]. Therefore, a reduction or stability in lesion size has long been considered a marker for a good treatment response. Nevertheless, recent evidence has shown that nearly 50% of BM may transiently enlarge after treatment without disease progression. For this reason, size alone is not able to provide a realistic estimate of treatment response [9]. Nor could post-contrast enhancement alone be considered a valid marker to discriminate between related changes and disease persistence; indeed, an increase in contrast enhancement following radiation treatment may be related both to progressive disease, pseudoprogression, and radiation necrosis (RN) [10]. Also, if the patient has received concomitant immunotherapy, the appearance of new lesions may not constitute progressive disease [7]. 

Therefore, conventional imaging alone, based on the evaluation of T2 signal intensity changes, contrast enhancement pattern changes, and size assessment alone, is not able to correctly identify treatment response and distinguish it from therapy-induced related changes [7,11]. Other techniques—some routinely used, others innovative—could help: restriction of diffusion in Diffusion Weighted Imaging (DWI) is usually a biomarker for hypercellularity [12]; dynamic susceptibility contrast-enhanced (DSC) MR perfusion imaging is usually a marker of neoangiogenesis and reduction of its derived parameter rCBV (relative Cerebral Blood Volume) is often seen after SRS [10]; and dynamic contrast-enhanced perfusion (DCE) as a tool to evaluate permeability changes [10,13]. 

In this manuscript, we revised the literature and reported the role of the main diagnostic techniques in helping neuro-oncologists and radiation therapists distinguish changes induced by treatments from disease progression. Since the radiation-induced changes are time-dependent, we described the role of both early and late MRI techniques in patients with BM after focal and systemic treatments; moreover, we tried to summarize how to interpret brain changes in patients receiving immunotherapy.

## 2. How to Translate in Imaging the Effects of Stereotactic Radiosurgery on Tumoral Cells and Surrounding Brain Cells

It has been widely reported that the risk of radiation injury correlates partly with lesions size and location, volume of normal brain parenchyma receiving radiation, radiation dose, prior use of RT, and concurrent systemic treatments including either immunotherapy or targeted therapy [14,15,16]. 

SRS achieves its therapeutic effects in a time- and dose-dependent way by causing DNA damage, resulting in the inhibition of tumor cell division, induction of apoptosis or necrosis, and thrombosis of neoplastic vessels [2,17]. Mechanisms underlying the effectiveness of SRS and consequently the brain SRS-related changes are multiple, not merely related to tumoral cell killing but also involving the tumoral microenvironment. The main mechanisms involved are summarized below (the first one mostly affects tumoral cells; the others affect the tumoral microenvironment).

DNA injury and apoptosis: ionizing radiation produces oxygen-free radicals in tumor cells, inducing cell death mainly due to the breakage of the DNA double helix; DNA repair pathways are subsequently activated, leading to cell cycle arrest and apoptosis of cells with irreversibly damaged DNA [18,19].Ceramide-induced apoptosis and fibrinoid necrosis: radiation directly damages the plasma membrane of several cell types (like endothelial cells), activating the enzymatic hydrolysis of sphingomyelin, which generates ceramide. Ceramide acts as a second messenger, stimulating ‘’ceramide-induced apoptosis’’ via the mitochondrial system [20]. This process leads to the production of more reactive oxygen species, which subsequently induce an inflammatory response involving cytokines and chemokines and then the formation of fibrin-platelet thrombi and fibrinoid necrosis [21].Demyelination and diffuse edema: astrocytes, oligodendrocytes, and neural progenitor cells are extremely sensitive to radiation, and radiation damage in the brain results in foci of demyelination. Moreover, necrotic tumor debris, if not readily removed, causes an inflammatory response that induces a capillary permeability defect with consequent edema. The preferential sites of this phenomenon are represented by basal nuclei, cerebral peduncles and deep white matter [22,23].HIF-1 and VEGF activation and neoangiogenesis: it has been demonstrated that radiation injury increases the release of HIF-1a and VEGF by astrocytes. The upregulation of HIF-1a leads to angiogenesis [24], with new fragile and leaking vessels causing perilesional edema.Blood–brain barrier (BBB) disruption: the disruption of the BBB caused by radiation leads to cerebral vasogenic edema. Radiation furthermore induces transient vasodilatation, with variable alteration of capillary permeability generally reversible and transient [25].

In the radiological field, all these phenomena translate, respectively, into enhancement after contrast medium injection, hyperintensity on T2-weighted images related to vasogenic edema, and necrotic areas.

## 3. Early Post-Treatment Assessment of Stereotactic Radiosurgery


*Key points*



*Increased diffusivity could be an early sign of radiation treatment efficacy.*

*The reduction of the rCBV DSC-derived parameter within the lesion has been generally considered a reference target for the effectiveness of RT.*

*The reduction of the K-trans DCE-derived parameters is related to a good response to treatment due to a reduction in the pathological vascular permeability of the treated area.*


In the early post-SRS phase (typically within three months after treatment), the main neuroradiological goal is to correctly interpret the RT effects in order to identify well-responding patients with important prognostic implications and unresponsive patients who may benefit from further treatments [9,26].

### 3.1. Diffusion Weighted Imaging (DWI)

DWI and the derived apparent diffusion coefficient (ADC) are based upon water molecule mobility in tissue and provide indirect information on the tissue microenvironment. The role of DWI as a biomarker in the detection of early post-radiation effects has been extensively studied [4,12,27]. Based on the principle that restricted diffusion is a marker of hypercellularity, it could be speculated that as cells are killed by therapy in the BM, this could lead to increased diffusivity, which could potentially be an early sign of radiation treatment efficacy [4,12,27]. In this regard, Huang et al. showed how the ADC value in BM increased significantly already in the first days after SRS [13], whereas Chen et al. identified the Diffusion Index (tumor volume/ADC mean) as a valid biomarker with lower Diffusion Index values one month after SRS in responder patients compared to non-responder patients [27].

### 3.2. Dynamic Susceptibility Contrast (DSC) Perfusion MRI

DSC perfusion imaging is the most widely used perfusion technique. It is suitable to be performed routinely in the follow-up of irradiated BM as it only takes about 2–3 min to be acquired and is highly sensitive in discriminating tumors from post-treatment changes [10,26]. 

DSC perfusion MRI relies on the T2 and T2* shortening effects of gadolinium-based contrast agents. It is performed using a series of T2*-weighted gradient echo-planar images acquired during the passage of a standard dose (0.1 mmol/kg) of contrast agent intravenously administered at a rate of at least 3 mL/s.

The main parameters derived from DSC MRI are the rCBV and the relative cerebral blood flow (rCBF), which represent, respectively, the volume of blood within the lesion and the volume of blood passing through the lesion per unit of time normalized to the contralateral normal parenchyma; those parameters are strictly related to tumoral neoangiogenesis. Other DSC parameters, derived from the T2* signal-intensity time curve, are the relative peak height (rPH) and the percentage of signal-intensity recovery (PSR).

A reduction in the rCBV parameter within the lesion has been generally considered a reference target for the effectiveness of RT, even in the early stages [4].

Although DSC theoretically can early discriminate responders from non-responders, it is important to remind about some issues concerning this technique, first of all the limit related to paramagnetic artifacts that in several cases hamper DSC in predicting tumor response after SRS; the difficulty of evaluating lesions in highly vascularized cortical areas; and the issue related to extravascular leakage of contrast agent [10]. 

### 3.3. Dynamic Contrast Enhanced (DCE) Perfusion MRI

An alternative, less commonly employed, perfusion technique is DCE MRI, which involves serial T1-weighted images before, during and after gadolinium-based contrast agent injection over a prolonged time of acquisition, typically 5 min or longer. DCE MRI is a multiparametric perfusion whose data analysis can be performed using both qualitative, semi-quantitative and quantitative methods in order to achieve information about tumor microvasculature and microarchitecture. Particularly, it assesses tumor microvasculature and is able to evaluate permeability changes within the treated lesion [13]. The transfer constant (Ktrans) is the most used parameter, reflecting flow and permeability [28]. Other permeability parameters include volume fraction of extracellular extravascular space (Ve), reflux rate (Kep), and vascular plasma volume (Vp). Ve depends on cellular density, tissue architecture and the presence of necrotic areas; Kep represents the reflux rate of gadolinium from the extracellular extravascular space back into plasma and depends on both Ktrans and Ve (Kep = Ktrans/Ve). Vp reflects the blood plasma volume per unit volume of tissue, and it is related to neoangiogenesis and vascular density [29].

Regarding the role of early-stage DCE perfusion, Taunk et al. have shown that lower values of the K-trans after SRS are related to a good response to treatment due to a reduction in the pathological vascular permeability of the treated area [30], and according to Knitter et al., an increase in the K-trans values, even in the early stage, is related to the progression of disease [4].

## 4. The Role of Neuroimaging in Distinguishing True Disease Progression from Post-Treatment Radiation Effects (PTRE) Mimicking Disease Progression


*Key points*



*Radionecrosis and pseudoprogression are possible post-SRS treatment changes.*

*An enhancing lesion may represent both tumor recurrence and post-treatment radiation effects; T1 mapping could help in differential diagnosis with continuous but slow accumulation of contrast agent in RN in contrast to the rapid contrast agent accumulation and relatively fast clearance in tumor recurrence.*

*In DWI/ADC images, “The centrally restricted diffusion sign” appeared to be due to hypercellularity in coagulative necrosis and theexpression of RN.*

*DSC helps in differentiating pseudoprogression, or RN, from progressive disease, with the highest value of rCBV in progressive disease.*

*Ktrans and Vp DCE-derived parameters seem to help in differentiating progressive disease from radiation injuries; anyway, the role of DCE is still debated in the literature.*

*ASL seems to be useful only in monitoring metastatic lesions characterized by high vascularity and increased CBF values, including renal cell carcinoma, melanoma and thyroid carcinoma.*

*PET imaging, with 18F-fluorodeoxyglucose or amino acid tracers, represents an additional tool. Typically, high uptake of tracers is observed in tumor recurrence, while low uptake is considered a hallmark of radiation effects.*

*Radiomics and AI are showing promising results in differentiating true progression from treatment effects, but they still must be validated.*


Post-SRS treatment radiological follow-up includes serial magnetic resonance imaging, usually at two- or three-month intervals. During the follow-up, the main goal is to differentiate true disease progression from its mimics induced by RT, including pseudoprogression and RN.

A correct and timely diagnosis is important due to their difference in management and survival. Tumor progression is managed with surgery or further RT, while post-radiation treatment effects are managed with observation, steroid therapy, or vascular endothelial growth factor inhibitors such as bevacizumab.

Pseudoprogression is radiologically defined as the appearance of a new area of enhancement or as the enlargement of a pre-existing lesion within the field of irradiation in the absence of true tumor growth, which subsides or stabilizes on follow-up imaging without a change in therapy [28]. It may occur in a time frame ranging from 6 weeks to around 15 months after SRS treatment [31]. This complex process seems largely due to a combination of post-radiation treatment effects such as endothelial cell injury and local inflammation, with increased vessel permeability leading to a new or enlarging area of post-contrast enhancement, and its reported incidence in BMs ranges from 9% to 30% [32].

RN represents the most severe case of radiation injury in the late stage, occurring typically 6–24 months after the end of radiotherapy. It is characterized by more extensive tissue and vascular damage, resulting in fibrinoid necrosis. Its incidence is estimated to be between 5 and 25% after RT, depending on the prescribed dose, volume of brain irradiated, previous use of radiation and concurrent systemic therapy. It appears on MRI as a contrast-enhancing necrotic lesion, surrounded by edema, localized within the field of irradiation, and, due to common features, hardly distinguishable from tumor recurrence. Enhancement patterns described in the literature as “Swiss cheese,” “soap bubble,” or “cut green pepper” have long been related to RN. However, nowadays, recent studies have demonstrated that they have only a 25% positive predictive value [33,34].

Realistically, in many cases, enhancing lesions are a combination of both tumor recurrence and post-treatment effects. Since distinguishing radiation injury from tumor progression is a difficult clinical dilemma routinely faced, several imaging techniques have been implemented to achieve the correct diagnosis. 

### 4.1. Post-Contrast T1-Weighted Sequences

A clinically dedicated MRI protocol f patients with BM should always include post-contrast T1-weighted sequences. These sequences highlight areas of altered blood–brain barrier characterized by enhancement post-administration of gadolinium-based contrast agents. As mentioned above, a gadolinium-enhancing lesion found on MRI imaging may represent both tumor recurrence and post-treatment radiation effects. 

To overcome this limit, Wang et al. investigated the efficiency of post-contrast T1 mapping in the differential diagnosis of RN and tumor recurrence in BM. T1-mapping images were acquired 5 min (T15min) and 60 min (T160min) after contrast injection, using a gamma knife rigid head frame to guarantee identical slice position and angulation. There were significant differences between the 2 groups in T15min, T160min, and T1differ (T160min–T15min) values (*p* = 0.012, *p* = 0.004, and *p* < 0.001, respectively). Theoretically, the T1differ value should be negative in RN and positive at tumor recurrence. It seems to be a consequence of a continuous but slow accumulation of contrast agent for damaged vessels in RN, in contrast to the rapid contrast agent accumulation and relatively fast clearance in tumor recurrence in the presence of a rich web of undamaged vessels [35]. Similar results were reported by Zach et al.; they described that the region of contrast accumulation in post-contrast T1-weighted images acquired 60 min later than contrast injection correlated to non-tumor tissues with 100% sensitivity and 92% positive predictive value [36].

### 4.2. Diffusion Weighted Imaging (DWI) 

Some studies noted that ADC values are usually higher in RN and pseudoprogression than in recurrence [4,33].

Shifting from a quantitative to a qualitative pattern-based approach, Hainc et al. [37] differentiated RN from tumor progression in SRS-treated BM using a visual interpretation of DWI. In 59 patients with BM showing ring-enhancing and central necrosis on follow-up MRI after SRS (mostly performed after at least three months from SRS), they found four different DWI patterns: central restricted diffusivity (“the centrally restricted diffusion sign”) (Figure 1)peripheral restricted diffusivityboth central and peripheralno diffusion restriction.

They found, with ground truth determined by histopathology, that the presence of “the centrally restricted diffusion sign”, exclusively within the ring-enhancing lesion or combined with peripheral diffusion restriction, significantly differentiates RN from tumor progression (*p* < 0.001, sensitivity of 83% for RN, positive predictive value of 59%, and negative predictive value of 85%). In the absence of the centrally restricted diffusion sign, the probability of RN was low. Histologically, the association between central diffusion restriction and RN appeared to be due to hypercellularity in coagulative necrosis as a result of hemosiderin-laden macrophages and fibroblastic and inflammatory cells proliferation [37].

### 4.3. Dynamic Susceptibility Contrast (DSC) Perfusion MRI

One of the most useful parameters to use to discriminate radiation-related changes from tumor persistence is rCBV. In fact, it increases tumor recurrence due to the presence of increased microvascular density, while it is lower in areas of pseudoprogression and RN. Indeed, several studies of DSC have reported good sensitivity and specificity of rCBV in differentiating psuedoprogression or RN from progressive disease [4,14,38]. 

Minniti et al. reported high accuracy in distinguishing tumor recurrence from post-treatment changes using the following MRI criteria as suggestive of post-treatment effects: reduced or stable lesion over a 4-month interval and reduced perfusion on dynamic MRI sequences, with an rCBV cut-off of <2.0 or a clear absence of perfusion (“black hole”), in the absence of any nodular highly vascularized area within the contrast-enhanced lesion at DSC perfusion MRI [14]. Cicone et al. identified the best differentiating rCBV cut-off value of 2.14 (accuracy 75.6%, sensitivity 86.7%, specificity 68.2%), supporting the study of Minniti et al. [38]. 

Other parameters that may be useful include relative peak height (rPH), maximum change in signal during the passage of contrast agent, and the percentage of signal-intensity recovery (PSR), an indicator of blood-brain-barrier integrity that reflects gadolinium extravascular leakage. Indeed, tumor recurrence typically shows relatively higher rPH and lower PSR. Higher rPH correlates with increased tumor capillary blood volume, while decreased PSR correlates with abnormally formed new vessels, which allow more gadolinium extravascular leakage, leading to a slow signal recovery and consequently to a persistently decreasing signal. In contrast, RN is characterized by significant vascular damage, which is represented not only by decreased rCBV and rCBF but also by lower rPH values and higher PSR values due to less leakage of contrast into the extracellular space [39].

Barajas et al. demonstrated that among DSC parameters, PSR was the best indicator of RN, reaching a sensitivity of 95.65% and a specificity of 100% with a cut-off value of >76.3% [40].

For DSC, the technical issues mentioned above remain. 

### 4.4. Dynamic Contrast Enhanced (DCE) Perfusion MRI 

In a pilot study, Hatzoglou et al. [41] demonstrated that DCE-MRI could be a valid tool to evaluate nonspecific enhancing intracranial lesions after RT. They found a higher Vp ratio (VP lesion/VP normal brain) and Ktrans ratio (Ktrans lesion/Ktrans normal brain) correlated significantly with progression. The Ktrans ratio and Vp ratio optimal thresholds to distinguish progression from radiation injury were estimated to be 3.6 and 2.1, respectively. In their study, the Vp ratio was found to be the most effective metric (AUC 0.87, sensitivity 92%, specificity 77%), with an accuracy increase to 94% if combined with the Ktrans ratio. 

If it is true that in the literature the Ktrans value tends to be higher in tumor recurrence than in radiation-induced changes, it is also true that in the various studies proposed Ktrans thresholds vary widely, with no currently established reproducible threshold. Furthermore, there is discordance in the literature about the role of the other DCE-derived parameters [41,42].

Indeed, one of the main limitations of DCE is its poor reproducibility, which mainly depends on scanners and post-processing software. Other limitations are the long acquisition time (>5 min), lack of expertise and intuitive post-processing software, which still limit its wide use. However, DCE may help when DSC is uninterpretable due to susceptibility artifacts, or it can be used in addition to DSC and the contrast medium injected to acquire DCE can result in a useful pre-bolus for DSC acquisition [43].

### 4.5. Arterial Spin Labeling (ASL) Perfusion MRI

In contrast to DSC and DCE imaging, ASL is a non-contrast perfusion method that is particularly useful in patients with poor renal function and difficult intravenous access after chemotherapy. ASL relies on magnetically labeled blood as an endogenous tracer to determine cerebral blood flow (CBF) [43].

The increased tumor metabolism in BM leads to an increase in tumor vascularity, and this translates into an increase in CBF value, which instead does not occur in RN. 

Lai et al. compared the diagnostic sensitivity and specificity of FDG-PET and ASL to detect tumor recurrence after SRS, using histology as the gold standard. FDG-PET and ASL were equally sensitive in detecting tumor progression (83%). However, the specificity of ASL was higher (100%, 75%, and 50%, respectively) [44]. 

The main limitation of the use of ASL is that it can only be useful in monitoring metastatic lesions characterized by high vascularity and increased CBF values, including renal cell carcinoma, melanoma and thyroid carcinoma, while its role in poorly vascularized metastases (i.e., lung and breast cancer) is still unclear. Additionally, CBF in previously irradiated tumor recurrence areas may be underestimated due to the delayed arrival of arterial blood for damaged vessels [44] (Figure 2).

### 4.6. Positron Emission Tomography (PET) Imaging

In the diagnostic challenge of differentiating post-radiation treatment effects from tumor recurrence, PET imaging with 18F-fluorodeoxyglucose or amino acid tracers may represent an additional investigation even if it is not routinely used in clinical practice due to limited availability, expensiveness, and long acquisition time.

PET utilizes the ability of brain tissue to uptake radiotracers. Typically, high uptake of tracers is observed in metabolically active lesions such as tumor recurrence, while low uptake is considered a hallmark of radiation effects because of the relative lack of vital tissue.

18F-fluorodeoxyglucose (FDG)-PET images are obtained by a PET/CT scanner about one hour after intravenous injection of FDG. The maximum standard uptake value (SUV) is calculated by drawing an ROI. Its sensitivity and specificity in discriminating disease progression and post-treatment-related effects for BM are reported to amount to, respectively, 86% and 80% [44]. 

Le Rhun et al. found that the sensitivity of FDG-PET and ASL was similar for the detection of tumor progression and radiation effects, estimated at 83%. However, the specificity of ASL was higher than that of FDG-PET (100% for ASL, 75% for FDG-PET) [45].

Hatzoglou et al. evaluated the efficacy of FDG PET-CT and DCE MRI in differentiating tumor progression and radiation damage in patients with indeterminate enhancing lesions after RT. When Ktrans ratio ≥ 3.6, Vp ratio ≥ 2.1, and SUV ratio ≥ 1.2 were combined, the rate of correct classification of progressive disease and radiation injury reached 79% and 94%, respectively. However, DCE-MRI performed better than PET-CT when results were discordant [41]. 

This data may have important implications for clinical practice since perfusion MRI techniques are less expensive and less time-consuming, and MRI is routinely used in follow-up.

There are limitations that can arise during the interpretation of FDG-PET, such as differentiation from normal cortical uptake. In this regard, amino acid tracers may be more suitable due to the high amino acid metabolism in the tumor and relatively lower uptake in normal brain tissue, providing a good signal-to-noise ratio [33].

Very encouraging results have been obtained using amino acid tracers, including 6-[18F]-fluoro-L-3,4-dihydroxyphenylalanine (F-DOPA), L-[methyl-11C] methionine (MET) and O-(2-18Ffluoroethyl)-L-tyrosine (FET) [34]. 

Several studies, focused on the differentiation among radiation injury and progression, report the diagnostic accuracy of F-DOPA, MET and FET to amount to approximately 80–90% [28,46,47].

Cicone et al. [38] compared F-DOPA PET and DSC MRI perfusion in 42 patients with BM following SRS, in differentiating RN from progressive disease. Using a cut-off value of 1.59 as the maximum lesion to maximum background uptake ratio (SUVLmax/Bkgrmax) for F-DOPA, they found that PET performed better than DSCMRI with a sensitivity of 90% and a specificity of 92.3%) [38]. A known limitation of F-DOPA in assessing brain tumor metabolism is the increase in physiological uptake of the basal ganglia, which may interfere with the tumor signal.

Of particular interest is FET-PET, whose dynamic uptake data analysis has shown a sensitivity of 100% and a specificity of 93%. In fact, RN shows a steadily increasing curve pattern, whereas progression disease has an early peak of amino acid uptake followed by either a plateau or a fast decline [6,46] (Figure 3 and Figure 4).

### 4.7. Radiomics and Artificial Intelligence

Radiomics and artificial intelligence (AI) studies in neuro-oncology are growing exponentially, with promising results for differentiating between treatment effect and true progression in BM treated with SRS [48]. Larroza et al. used texture analysis and a machine-learning classification technique to differentiate between BM and RN based on contrast-enhanced T1-weighted images, achieving high classification accuracy (AUC > 0.9) [49]. Another interesting aspect is the role of radiomics and, above all, of artificial intelligence in stratifying BM radiosensitivity to SRS [50].

Despite these promising results, validation with large multicenter and heterogeneous datasets is needed to confirm the performance accuracy of radiomics and AI methods before deployment in the clinical neuro-oncology setting.

## 5. The Impact of Systemic Treatment in Neuroimaging Changes


*Key points*



*Immunotherapy leads to new challenges in the imaging interpretation of post-SRS BM, with a high rate of post-treatment changes, especially pseudoprogression.*


The introduction of immunotherapy for BM has drastically improved the prognosis and overall survival of patients. The best results were found in patients with melanoma, lung cancer and renal cell carcinomas. The goal of these new drugs is to block regulatory checkpoints of the immune system, such as cytotoxic T lymphocyte-associated protein 4 (CTLA-4) and cell death receptor 1 (PD1) axis and allow the immune system to attack cancer cells [51,52,53,54].

The combination of radiotherapy with immunotherapy can have both local and distant synergistic effects: the release of tumor antigens following neoplastic cell death activates the cytotoxic immune response against the remaining tumor cells, with effects also at a distance from the site of irradiation (abscopal effect) [55,56,57]. Numerous retrospective studies have confirmed the superiority of immunotherapy treatments or target therapy combined with radiosurgery compared to monotherapy [58,59].

Nevertheless, Colaco et al. reported an increased rate of treatment-related imaging changes in patients receiving immunotherapy compared with those who receive chemotherapy or targeted therapy after SRS [60].

A recent literature systemic review has evaluated the efficacy and safety of SRS alone or combined with immunotherapy for the treatment of melanoma BM. By analyzing four studies including 367 patients, the authors found that combined treatment is associated with better brain control and longer survival without an increased risk of radiation-induced toxicity profile; however, current evidence remains low due to the absence of randomized trials [61]. The introduction of these new therapeutic strategies has led to new issues in the interpretation of MRI, with pseudoprogression becoming more frequent in patients treated with immunotherapy. Again, pseudoprogression represents a real challenge for clinicians and neuroradiologists, as the lack of recognition of these entities can lead to a premature interruption of successful therapies [57,62,63,64].

Size assessment cannot be sufficient in the follow-up evaluation of patients treated with SRS and immunotherapy. Indeed, the problem of two-dimensional and volumetric measurements in patients treated with immunotherapy is related to the presence of an inflammatory infiltrate that can mimic disease progression in the early post-treatment stages [64].

DWI also may be a confounding factor in patients treated with immunotherapy; if usually a decrease in ADC value is correlated with disease progression [65,66], inflammatory infiltrate can cause a restriction of diffusion, simulating disease progression. Currently, no studies have ADC values in patients with BM treated with immunotherapy alone versus combined SRS and immunotherapy [64]. 

In this context, DCE-MRI and DSC-MRI perfusion techniques play an important role in the definition of disease progression, given that the increase in blood flow should correlate with the presence of an active tumor, even in patients undergoing combined treatments [64,67,68]. In particular, the absence of an increase in the rCBV-DSC-derived parameter at the sites of pathological enhancement seems to suggest a treatment-related effect rather than a disease progression in patients treated with immunotherapy [69]. However, further studies are needed to determine the effect of immunotherapy on the molecular substrate and tumor microenvironment and how this affects magnetic resonance imaging, taking into account that BMs have different biological behaviors based on the origin of the primary tumor [64].

## 6. Conclusions

In conclusion, RT causes a series of pathophysiological changes in brain tissues that make it challenging to assess the response to therapy. In the early stages, it is important to identify patient responses to the therapy. In the following imaging studies, it is crucial to differentiate progression disease from pseudoprogression, or RN. Both may mimic disease on conventional MRI images, so neuroradiologists should go beyond the dimensional data and properly use all the available MRI techniques, including the newest one, in order to give neuro-oncologists and radiotherapists information on the disease status.

## Figures and Tables

**Figure 1 cancers-15-05092-f001:**
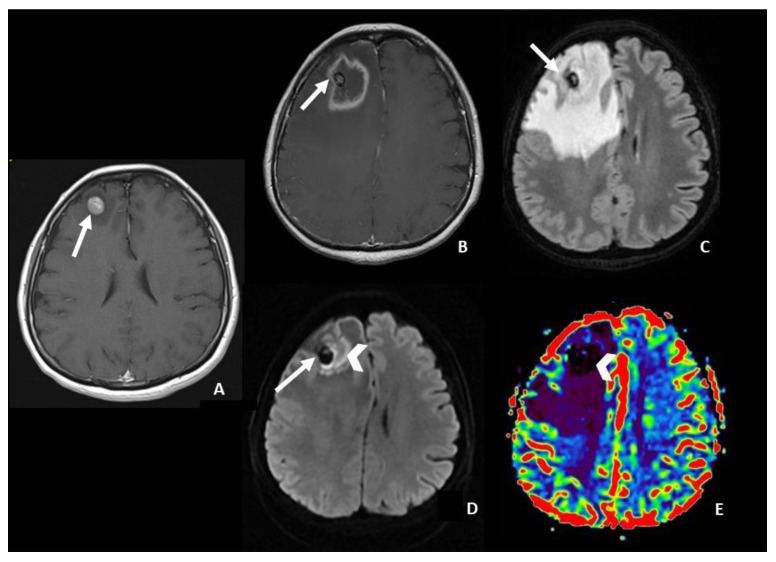
Right frontal brain metastasis in a patient affected by lung cancer. (**A**) Axial post-contrast T1 WI showing the brain metastasis before SRS (arrow). (**B**–**E**) Follow-up MRI 8 months after SRS showed on axial post-contrast T1 (**B**) a volumetric increase of the lesion that is characterized by rim enhancement, necrotic core with an associated central hemorrhage (arrow, on (**B**–**D**)), and abundant perilesional edema on axial FLAIR (**C**). The presence of coagulative necrosis characterized by diffusion restriction on DWI (arrowhead on (**D**)) and the absence of increased r-CVB lesional values (arrowhead on DSC rCBVmap (**E**)) lead to radiation necrosis changes. WI = weighted image; SRS = Stereotactic radiosurgery; FLAIR = fluid-attenuated inversion recovery; DWI = diffusion-weighted imaging; rCBV = relative Cerebral Blood Volume; DSC = Dynamic susceptibility contrast-enhanced.

**Figure 2 cancers-15-05092-f002:**
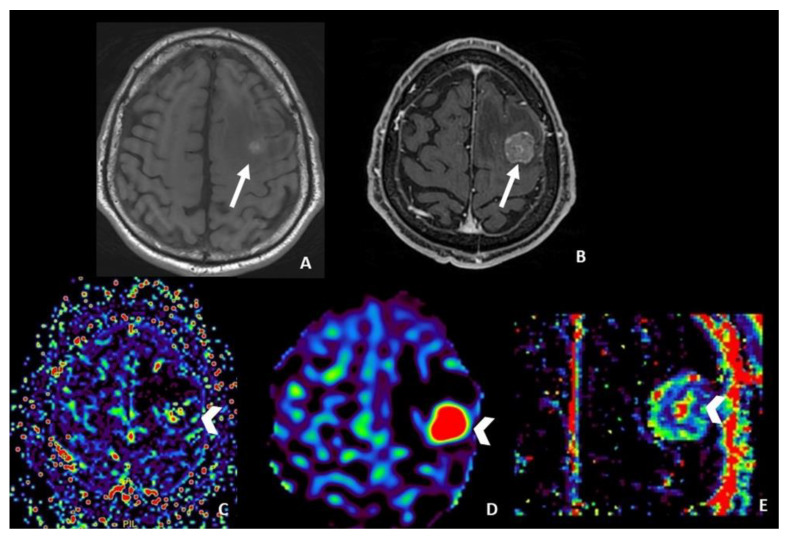
MRI follow-up 6 months after SRS in a patient affected by melanoma with left frontal partial hemorrhagic ((**A**) pre-contrast T1 WI) brain metastasis ((**B**) post-contrast T1 WI). DSC-derived rCBV map ((**C**) arrowhead) showing doubtful increased values, difficult to interpret because of the presence of blood material. ASL-derived CBF map ((**D**) arrowhead) and DCE-derived Ktrans map ((**E**) arrowhead) show a clear increase of those parameters compatible with the presence of disease. WI = weighted image; SRS = Stereotactic radiosurgery; rCBV = relative Cerebral Blood Volume; DSC = Dynamic susceptibility contrast-enhanced; CBF = cerebral blood flow; ASL = Arterial spin labeling; DCE = Dynamic contrast enhanced; Ktrans = transfer constant.

**Figure 3 cancers-15-05092-f003:**
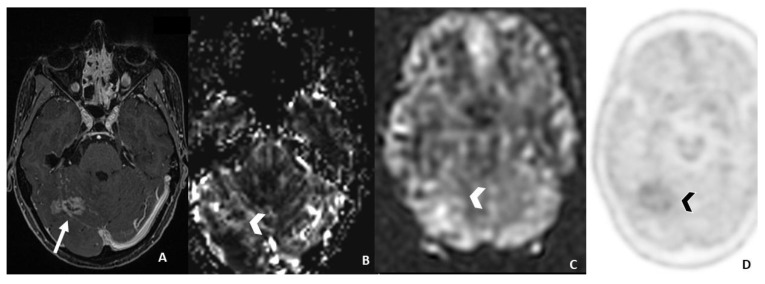
MRI follow-up three years after SRS in a patient affected by breast cancer with brain metastasis, still showing an enhancing area ((**A**) post-contrast T1 WI) brain metastasis, without increased DSC-derived rCBV ((**B**), arrowhead) and ASL-derived CBF ((**C**), arrowhead) values and without increased uptake on 18-F-DOPA PET/TC) ((**D**), arrowhead). These findings lead to radiation-induced changes. WI = weighted image; SRS= Stereotactic radiosurgery; rCBV = relative Cerebral Blood Volume; DSC = Dynamic susceptibility contrast-enhanced; CBF = cerebral blood flow; ASL = Arterial spin labeling; DCE = Dynamic contrast enhanced; Ktrans = transfer constant; F-DOPA = dihydroxyphenylalanine.

**Figure 4 cancers-15-05092-f004:**
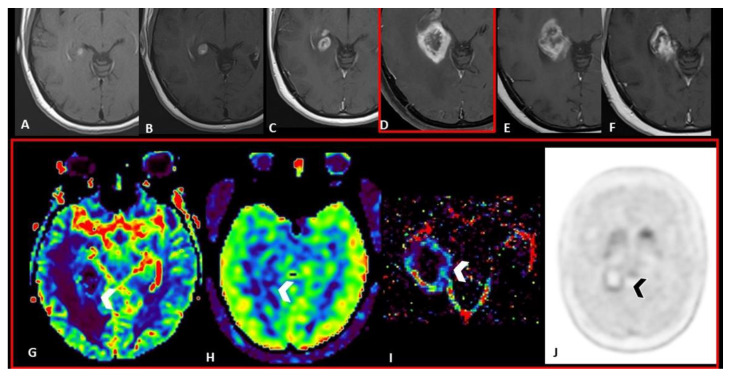
Longitudinal MRI follow-up in a Patient affected by lung cancer with brain metastasis. Axial post-contrast T1 WI at time zero (**A**), and, respectively three (**B**), five (**C**), seven (**D**), nine (**E**), and eleven (**F**) months after SRS showing a progressive lesional increase in terms of volume and enhancement from time zero to the seventh month; followed by a progressive volumetric reduction from the seventh to the eleventh month. MRI perfusion techniques in the seventh month do not show a significant increase in DSC-rCBV ((**G**), arrowhead), ASL-CBF ((**H**) arrowhead), and DCE-Ktrans ((**I**) arrowhead) values. F-DOPA PET/TC ((**J**) arrowhead) acquired in the seventh month does not show a significant uptake. These findings are compatible with pseudoprogression. WI = weighted image; SRS = Stereotactic radiosurgery; rCBV = relative Cerebral Blood Volume; DSC = Dynamic susceptibility contrast-enhanced; CBF = cerebral blood flow; ASL = Arterial spin labeling; DCE = Dynamic contrast enhanced; Ktrans = transfer constant; F-DOPA = dihydroxyphenylalanine.

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
