# Peer review of "Radiosurgery for Brain Metastases: Challenges in Imaging Interpretation after Treatment"

_cancers, 2023, doi:10.3390/cancers15205092_

Round 1

Reviewer 1 Report

The authors tried to mention every topic in their review.

At the end it is very difficult for the reader to get the main important points.

Therefore, I clearly recommend to restructure the review.

eg:

early (changes up to three months) and their imaging feature description, images; short important results summary.

late changes,...short important results summary

progression, pseudoprogressiion, pseudoresponse need exact definitions and then image descriptions. short important results summary

further remarks:

l.108: leads

ll: 144: wrong definition

l 151: other issues, which more?

l 307 DSC and DCE not universal, exactly define "universal"

322: Do you mean: histology

325: ASL

379 : "d"?

439/3440: Incomplete

473: Do you mean: conventional images?

Moderate corrections required.

Author Response

Dear Editors

Dear Reviewer

Thank you for your precious comments and suggestions.

Particularly, we tried to make the review more readable and useful by adding to the main paragraphs few key points.

Here listed (in red) the responses to your comments. Also, any changes in the main text have been highlighted.

1st Reviewers

The authors tried to mention every topic in their review.

At the end it is very difficult for the reader to get the main important points.

Therefore, I clearly recommend to restructure the review.

eg:

early (changes up to three months) and their imaging feature description, images; short important results summary.

late changes,...short important results summary

progression, pseudoprogressiion, pseudoresponse need exact definitions and then image descriptions. short important results summary

R: thank you for your comment, we modified the text adding a short list of key points for the more complex paragraphers

further remarks:

l.108: leads

R thank you, modified

ll: 144: wrong definition

R thank you, modified

l 151: other issues, which more?

R other issues added

l 307 DSC and DCE not universal, exactly define "universal"

“universal” chanced with “reproducible.”

322: Do you mean: histology

Yes, modified.

325: ASL

Thank you modified.

379 : "d"?

Thank you, removed.

439/3440: Incomplete

R: data added

473: Do you mean: conventional images?

R: yes, modified

Reviewer 2 Report

This article mainly evaluates the imaging evaluation of brain metastases after radiation therapy, which has guiding significance of the problems faced in clinical work. It can help us distinguish between pseudoprogression and progression, which is of great significance for clinical treatment. The structure of the article is reasonable. Much of this article is devoted to the functions of MR and PET. There still has a small problem will be pointed out:  whether CT and angiography could help us in differentiating brain metastases progression?

Author Response

R: thank you for your comment, we did not add any information about CT and angiography because there are no literature data indicating these techniques as helpful tools in differentiating post-treatment changes from disease presence in treated brain metastases.

Reviewer 3 Report

This a review on the use of imaging after stereotactic radiosurgery of brain metastases. Early discrimination of recurrent tumor from radiation effects in the brain is clearly crucial for proper patient management. As noted by the manuscript, a lot of research has been done on identifying imaging techniques that can be useful for this discrimination. In that sense, this article touches on a very important topic for the neuro-oncology community. While the article touches on what I think are most the relevant imaging approaches, it can still be improved.

1.       Page 2 Line 51 to 53. I will point out that these two sentences do not correspond to the findings of the paper cited (citation 9). Rather, this is something that is mentioned in the introduction of that paper. You should instead refer to the original study which is Patel et al AJNR Am J Neuroradiol. 2011;32(10):1885–1892.

2.       Page 2 line 80. This paragraph and the subsequent bullet list are not as clear as it could be. The paragraph is more focused on the effect of radiation on the tumor while the bulleted list (beyond the first point) is about the effect of radiation on the surrounding brain tissue. Some clarity may be needed here so as to point out which mechanisms apply to the tumor and which to the normal tissue.

3.       Page 3 line 97. Citation 21 does not relate to fibrinoid necrosis. Neither word shows up on that paper. Instead, you could cite Perry, A. & Schmidt, R. E. Cancer therapy-associated CNS neuropathology: an update and review of the literature. Acta Neuropathol 111, 197–212 (2006). But there are multiple papers describing the appearance of fibrinoid necrosis after brain irradiation. Citation 21 is also referenced in the subsequent bullet point where it is actually relevant as that paper looks at the sensitivity of particular cell subpopulations.

4.       Page 3 line 108 to 11. For consistency with the rest of the list, it would be nice to have a citation here.

5.       Page 3 line 125. The wording of “could be speculated that BM reduce their cellularity upon response to therapy” makes it read as if the BM is choosing how it responds to radiation. It may be better to rephrase into something like “could be speculated that as cells are killed by therapy in the BM this could lead to increased diffusivity with could potentially be an early…”

6.       Page 4 Line 148 and 149. Why is this a 1-sentence paragraph? Citation 10 is inconsistent with this point if we focus on early response. They evaluated their patients at 3 to 12 months post irradiation. At 3 months, decreased rCBV was seen in 52% of their cases but was not predictive of the actual response.

7.        Page 4 Line 157. DCE MRI is not restricted to just Spin Echo images, it just needs to be T1-weighted.

8.       Page 5 Line 193. Citation 27 does not speak to the incidence of pseudoprogression being between 9 and 30%. The word pseudoprogression does not even appear on this paper.

9.       Page 5 Lines 207 to 208. This last sentence of the paragraph is unnecessary as it just lists what will be the following subsections. Again the citation 34 is mentioned but this  refers to something that appears in the introduction of the paper but is not the finding of said paper.

10.   Page 5 Lines 207 to 208. Where did citation 32 and 33 go? I see citation 33 in the next paragraph but citation 32 is not mentioned anywhere. I am assuming you had a section on the use of MRS which you removed at some point. There is value in bringing that section back,

11.   Page 5 Line 215 to 225. I wanted to point out Zach, L. et al. Delayed contrast extravasation MRI: a new paradigm in neuro-oncology. Neuro-Oncology 17, 457–465 (2015). This paper has a similar finding to citation 33 except it uses T1 weighted images instead of T1 maps.

12.   In general Section 3 and Section 4 of this manuscript, read as if they were written by separate authors. There is a very different level of methodological detail presented for the papers being cited. There is also some redundancy in these sections. Given that DWI, DSC, and DCE are already covered in Part 3, there doesn’t seem to be a need to go over why these techniques are effective again in Part 4. There needs to be some consolidation to bring these sections into alignment with each other and remove the redundant explanations.

13.   Page 5 line 232. I strongly disagree with the authors that a qualitative assessment is more efficient than a quantitative one, particularly when both are generated from exactly the same data. But I do conceive that there may be value in the pattern of enhancement. This also feeds into the radiomics section later in the paper which would seek to quantify these texture features.

14.    Page 7 line 301 to 304. There are no citations for these statements?

15.   Page 7 line 308. “< 5 minutes” should be “>”

16.   Page 7 line 313 to 320. There are no citations for these statements? CBF, as measured by ASL MRI, is not an “expression of the metabolic vascular coupling”. The description used in line 326 is more apt: increased ASL MRI means there is high vascularity.

17.   Page 9 Line 375. The sentences talks about F-DOPA PET. Citation 43 is about 11-C Methionine, citation 44 is about FET, and citation 45 does not include any PET data.

18.   Page 10 Line 426 and 431. I don’t think this paragraph is relevant. The reason immunotherapy has brought a lot of attention is because inflammation and the immune system can have an effect on the response of the normal tissue to radiation or an effect on the imaging response. Targeted therapies affect neither.

19.   Page 11 Line 439 and 44. The are words missing in this paragraph as depicted by the presence of ellipses.

20.   For section 5, you should have at least a few sentences on the effect of immunotherapy on the risk of developing RN. See for example: Colaco, R. J., Martin, P., Kluger, H. M., Yu, J. B. & Chiang, V. L. Does immunotherapy increase the rate of radiation necrosis after radiosurgical treatment of brain metastases? Journal of Neurosurgery 125, 17–23 (2016).

Author Response

Dear Reviewer

Thank you for your precious comments and suggestions.

Here are the responses to your comments. Also, any changes in the main text have been highlighted.

  1. Page 2 Line 51 to 53. I will point out that these two sentences do not correspond to the findings of the paper cited (citation 9). Rather, this is something that is mentioned in the introduction of that paper. You should instead refer to the original study which is Patel et al AJNR Am J Neuroradiol. 2011;32(10):1885–1892.

R: reference modified

  1. Page 2 line 80. This paragraph and the subsequent bullet list are not as clear as it could be. The paragraph is more focused on the effect of radiation on the tumor while the bulleted list (beyond the first point) is about the effect of radiation on the surrounding brain tissue. Some clarity may be needed here so as to point out which mechanisms apply to the tumor and which to the normal tissue.

R: thank you for your comment, the paragrapher has been slight modified in order to clarify which mechanisms are related to tumor cells and which are related to surrounding brain tissue.

  1. Page 3 line 97. Citation 21 does not relate to fibrinoid necrosis. Neither word shows up on that paper. Instead, you could cite Perry, A. & Schmidt, R. E. Cancer therapy-associated CNS neuropathology: an update and review of the literature. Acta Neuropathol 111, 197–212 (2006). But there are multiple papers describing the appearance of fibrinoid necrosis after brain irradiation. Citation 21 is also referenced in the subsequent bullet point where it is actually relevant as that paper looks at the sensitivity of particular cell subpopulations.

R: reference modified

  1. Page 3 line 108 to 11. For consistency with the rest of the list, it would be nice to have a citation here.

R: reference added

  1. Page 3 line 125. The wording of “could be speculated that BM reduce their cellularity upon response to therapy” makes it read as if the BM is choosing how it responds to radiation. It may be better to rephrase into something like “could be speculated that as cells are killed by therapy in the BM this could lead to increased diffusivity with could potentially be an early…”

R: modified

  1. Page 4 Line 148 and 149. Why is this a 1-sentence paragraph? Citation 10 is inconsistent with this point if we focus on early response. They evaluated their patients at 3 to 12 months post irradiation. At 3 months, decreased rCBV was seen in 52% of their cases but was not predictive of the actual response.

R: citation 10 removed from this sentence

  1. Page 4 Line 157. DCE MRI is not restricted to just Spin Echo images, it just needs to be T1-weighted.

R: modified

  1. Page 5 Line 193. Citation 27 does not speak to the incidence of pseudoprogression being between 9 and 30%. The word pseudoprogression does not even appear on this paper.

R sorry, there was a mistake with references order. The right reference has been added.

  1. Page 5 Lines 207 to 208. This last sentence of the paragraph is unnecessary as it just lists what will be the following subsections. Again the citation 34 is mentioned but this  refers to something that appears in the introduction of the paper but is not the finding of said paper.
  2. Modified
  3. Page 5 Lines 207 to 208. Where did citation 32 and 33 go? I see citation 33 in the next paragraph but citation 32 is not mentioned anywhere. I am assuming you had a section on the use of MRS which you removed at some point. There is value in bringing that section back,

R modified

  1. Page 5 Line 215 to 225. I wanted to point out Zach, L. et al. Delayed contrast extravasation MRI: a new paradigm in neuro-oncology. Neuro-Oncology 17, 457–465 (2015). This paper has a similar finding to citation 33 except it uses T1 weighted images instead of T1 maps.

R: Thank you, we added a sentence about this paper

  1. In general Section 3 and Section 4 of this manuscript, read as if they were written by separate authors. There is a very different level of methodological detail presented for the papers being cited. There is also some redundancy in these sections. Given that DWI, DSC, and DCE are already covered in Part 3, there doesn’t seem to be a need to go over why these techniques are effective again in Part 4. There needs to be some consolidation to bring these sections into alignment with each other and remove the redundant explanations.

R: We slight modified the paragraphers in order to achieve what required

  1. Page 5 line 232. I strongly disagree with the authors that a qualitative assessment is more efficient than a quantitative one, particularly when both are generated from exactly the same data. But I do conceive that there may be value in the pattern of enhancement. This also feeds into the radiomics section later in the paper which would seek to quantify these texture features.

R: We modified the sentence, the aim is to describe a qualitative approach to evaluate dwi based on the recognition of a pattern

  1. Page 7 line 301 to 304. There are no citations for these statements?

R: References added

  1. Page 7 line 308. “< 5 minutes” should be “>”

R thank you, modified

  1. Page 7 line 313 to 320. There are no citations for these statements? CBF, as measured by ASL MRI, is not an “expression of the metabolic vascular coupling”. The description used in line 326 is more apt: increased ASL MRI means there is high vascularity.

R: reference added. Sentence modified.

  1. Page 9 Line 375. The sentences talks about F-DOPA PET. Citation 43 is about 11-C Methionine, citation 44 is about FET, and citation 45 does not include any PET data.

R: Modified

  1. Page 10 Line 426 and 431. I don’t think this paragraph is relevant. The reason immunotherapy has brought a lot of attention is because inflammation and the immune system can have an effect on the response of the normal tissue to radiation or an effect on the imaging response. Targeted therapies affect neither.

R: Removed

  1. Page 11 Line 439 and 44. The are words missing in this paragraph as depicted by the presence of ellipses.

R: missing words added

  1. For section 5, you should have at least a few sentences on the effect of immunotherapy on the risk of developing RN. See for example: Colaco, R. J., Martin, P., Kluger, H. M., Yu, J. B. & Chiang, V. L. Does immunotherapy increase the rate of radiation necrosis after radiosurgical treatment of brain metastases? Journal of Neurosurgery 125, 17–23 (2016).

R: Added

Round 2

Reviewer 3 Report

All of my prior concerns have been addressed